# A multidimensional momentum chain model for tennis matches based on difference equations

Jingya Wang[1]*, Sihang Guo[2], Yuanyun Zhou[2]

1 School of Mechanical Engineering and Automation, Harbin Institute of Technology, Shenzhen, Shenzhen, Guangdong, China, 2 School of Computer Science and Technology, Harbin Institute of Technology, Shenzhen, Shenzhen, Guangdong, China

* 2292546744@qq.com

**Data Availability Statement:** All relevant data are available from https://github.com/JeffSackmann/tennis_slam_pointbypoint.

**Funding:** The author(s) received no specific funding for this work.

## Abstract

In the process of pushing the limits of human performance, competitive sports are dedicated to the pursuit of excellence. In this context, the concept of "momentum" has gained significant attention, as it is widely acknowledged to influence the outcomes of competitions. The question of whether momentum affects sports psychology and the mechanisms underlying its generation and influence merits thorough investigation. In this paper, taking the 7,284 scoring points in the men's singles tennis match at Wimbledon 2023 as an example, we expand upon traditional momentum research by integrating diverse algorithms, including statistical analysis and linear weighting, to construct a multidimensional momentum chain model predicated on difference equations, which aims to quantify the momentum dynamics for athletes in a match. To enhance the authenticity of our model, we incorporate a forgetting curve to modulate the momentum fluctuations. The results show that dominant players have significantly shorter running distances and higher success rates in net strokes than disadvantaged players, indicating that positive events markedly enhance players' psychological and behavioral performance. Furthermore, the likelihood of scoring is substantially greater for players possessing higher momentum, with data suggesting that the serving side has an 84% chance of securing a match victory. When applied to 6,870 tennis matches, our model achieves a prediction accuracy exceeding 80%. Accordingly, we have proposed tennis training suggestions based on the mechanisms of momentum and developed strategies to effectively harness the "hot hand" phenomenon in matches.

## 1. Introduction

The results of sports competitions and their match flow have always attracted the attention of athletes, coaches and audiences. The unexpected victories and defeats on the play field often provoke the exploration of the reasons behind them. Momentum, one of the most frequently mentioned and least understood phenomenon in sports, is often described as an unpredictable and invisible force that cannot be controlled by individuals or teams [1]. Many athletes,

**Competing interests:** The authors have declared that no competing interests exist.

coaches, and authorities believe that momentum can have a non-negligible impact on sports competitions.

The mechanism of momentum in sport has been a key focus target for a wide range of researchers since the 1970s. Richard Alderman [2] depicted the phenomenon of momentum systematically in a book entitled Psychological Behavior in Sport. Adler and Adler [3] then described momentum as a rising state of strength or intensity that can add fuel toward a goal, build up confidence of athletes and increase their likelihood of subsequent success. With regard to the generation and function of momentum, in early research there were two main categories: psychological momentum and behavioral momentum. On the one hand, many researchers believe that success is inseparable from an athlete's psychological state, among them Iso Ahola and Mobily [4] were the first to use the term psychological momentum. They argued that initial success may affect a person's self-confidence and thus improve his or her psychological functioning (i.e., attention, concentration, and thinking). On the other hand, some researchers insisted that it is physiology which plays a key role in momentum generation and functioning. Taylor and Demick [5] argued that physiological arousal is the most critical factor in momentum establishment, maintenance, and disruption, i.e., a contingency event that may lead to changes in various cognitive and physiological variables. At the same time, changes in momentum can elicit physiological feedback, including perception of situational control, personal-ability, self-efficacy, motivation, attention to task-related information, and physiological indicators (i.e., heart rate, respiration, and adrenaline). Both types of mainstream views are supported by evidence, for example, Iso Ahola and Mobily [4] verified the theoretical validity of their theory by measuring the link between initial success and the likelihood of subsequent success in squash matches. Additionally, the concept of behavioral momentum is also consistent with physiological concepts of arousal and anxiety associated with athletic performance [6, 7]. With further research on momentum, the concept of momentum deepened gradually, where psychological and behavioral changes were merged into one, and it was proposed that both psychological momentum (PM) [8] and behavioral momentum (BM) correspond to the driving force that is expected to bring about changes in performance. They represent both the psychological and behavioral aspects of the same phenomenon, referred to as psycho-behavioral momentum (PBM) [9], which reflects a composite phenomenon combining psychological, physiological and behavioral constructs.

After the concept of momentum was developed and refined, researchers turned to explore its role in sports, and one of the most frequently mentioned phenomenon was the "hot hand" phenomenon. Initially, the term was coined to describe the basketball players who, along with their supporters, believed that successfully scoring consecutive points, such as baskets, would enhance their probability of continuing that scoring streak. Conceptually, there appear to be many similarities between momentum and the mechanism of the "hot hand" phenomenon, and subsequent research has generally concluded that momentum and "hot hands" are the same phenomenon [9]. Therefore, in many studies, the existence of the "hot hand" phenomenon has demonstrated the exact effect of momentum on athletes' performance. Similarly, this study also considers the "hot hand" phenomenon and momentum to be the same concepts.

One of the prevalent psychological studies on the mechanism of momentum is self-efficacy theory. Self-efficacy theory posits that self-efficacy mediates the process by which success breeds success [10]. In sport, efficacy can be regarded as the ability to produce success (e.g., scoring). Successful performances and experiences are thought to be essential in fostering self-efficacy, while self-efficacy, in turn, promotes subsequent positive behaviors [11]. There has been many studies on the existence of self-efficacy theory in ball games. On the one hand, Avugos's study found that increased self-efficacy did not lead to higher performance in controlled shooting experiments conducted by professional and recreational basketball players [12]; On

the other hand, supportive evidence has been shown in individual sports (e.g., horseshoe bowling, bowling, tennis). It has been suggested that the function of self-efficacy may be largely influenced by team strategy as well as competition characteristics, so in this study, we will also explore the validity of this theory in depth for the individual sport of tennis.

Many studies on the physiological effects of momentum have shown a close relationship between momentum and hormones. In earlier studies, relationships between hormones and momentum were found in animals in a phenomenon known as the "winner effect". The winner effect is a theoretical term used to represent a variety of physiological, behavioral, and psychological responses to success or failure in insects [13], fish [14], reptiles [15], mammals [16], and nonhuman primates [17], and is behaviorally manifested in the fact that the animal that wins a battle is more likely to win the next battle. It has been found that hormones secreted endogenously following intense socialization experiences are the primary mediators of the winner effect, whereby victory increases testosterone levels and defeat dramatically decreases testosterone levels [18]. In the human domain, testosterone modulates behavior under competitive conditions [19]. Cheng and Kornienko [20] suggest that testosterone is a hormone that regulates the psychological system to adapt to the current social status and can direct competitive effort. Testosterone is effective throughout an athlete's competition and its effects are further enhanced when an athlete experiences high levels of success, such as winning a competition [21]. Mammals, especially humans, can spread winning signals to their surroundings, and high levels of testosterone help to dominate nonverbal communication, for example, through upright posture, strutting and confident facial gestures. Oppositely, low levels of testosterone inhibit these behaviors, leading to stooped postures, disgusted glances, submissive speech, and other different low-status gestures [22]. This mechanism of transmission (by the performer) and reception (by the environment) of success provides the biological basis for the discovery of momentum by observers [23].

From the above studies, it is clear that momentum has been researched with some depth and breadth in the field of human sport competition, and that positive momentum allows athletes to have higher levels of energy when performing tasks [24], as well as encouraging athletes to perform more powerfully in teams and to attempt more risky and proactive choices. Also, in response to changes in momentum, coaches and athletes tend to take timely measures to capitalize on or inhibit the effects of momentum. In Raab et al.'s study [25], volleyball players and coaches were able to perceive momentum and use it in tactical and strategic decisions, such as athletes perceived to have higher momentum were typically defended more tightly by the opposing team [26], and coaches used time-outs to stop other team's momentum from rocketing in a game, etc. [27]. Therefore, how to test the existence of momentum and measure the magnitude of momentum has become a key issue in sports competitions. For the detection and measurement of momentum, the earlier model theory comes from Taylor and Demick [5], who proposed a multidimensional momentum chain model that qualitatively describes the relationship between momentum and the outcome of the competition. The model consists of six main components, i.e., (a) contingencies, (b) cognitive, affective, and physiological changes, (c) changes in behaviors, (d) changes in performances that are consistent with the above changes, (e) successive and opposite changes in factors prior to the opponent, and (f) changes in outcomes as a direct result. The model provides a foundation for studying the effects and mechanisms of momentum in sports competitions. With the further development of the era, more and more studies have begun to focus on specific data differences and try to quantitatively describe the existence and role of momentum. For example, in the study of Page and Coates [28], they collected data from tennis matches in which the last two sets both ended in a tie, and found through statistical analyses that players who won the extra two points in a tie-break in the first set had a greater chance of winning the second set (60% versus 40%). At

the same time, this mathematical law appeared only in men's tennis matches, which provides testosterone's mediating mechanism on momentum with strong evidence. Though these two points represent only a small fraction of the total score for that handicap, Page and Coates insisted that the very act of winning or losing can affect the chances of further wins or losses. However, in the above study, they only considered the effect of winning or losing a tie-break on the subsequent game, and did not propose a momentum quantification method that can be applied to other game scenarios, so further research is needed. In the elite archery tournament, Yangqing Zhao a, Hui Zhang [29] innovatively proposed a method to quantify momentum. They used data analysis methods such as logistic regression and multiple linear regression to combine all kinds of variables in the tournament, and then constructed a measure of "heat" (i.e., momentum). Using this quantitative model, they provided an accurate and reliable basis for research and outcome prediction of archery competitions. Inspired by similar methods, this study attempts to apply statistical methods to tennis matches, and further provide methods for the discovery and quantification of momentum in tennis matches.

The purpose of this study is to investigate the "hot hand" effect in tennis, to validate the theories about the mechanism of momentum in tennis competitions and to explore other potential effects of momentum on human psychology and behavior, additionally, to propose a method to quantify momentum. Through this study, we have constructed a quantitative model for the momentum fluctuation of players in tennis, which not only proves the existence of the "hot hand effect" and the influence of momentum in tennis, but also provides a reliable basis for athletes and coaches to cope with different situations in different courts.

The main content of this study is as follows:

1. As this study mainly focuses on the "hot hand" effect in tennis competitions, we collected 7,284 data from the 2023 Wimbledon Men's Singles Tennis Championships and organized them into a standard dataset, and in the subsequent research and validation, we collected data from a total of 6,870 tennis matches from 2011 to 2023 (including Wimbledon, French Open, US Open and Australian Open), in order to ensure the data as comprehensive and objective as possible;

2. Momentum has always been regarded as an unknown and difficult to grasp force, in order to study the factors that cause changes in momentum in a tennis match, we used random forests and a variety of statistical analysis methods to carry out a statistical study on a number of the most common events in tennis matches, and found the events that were most associated with the probability of scoring points and winning, including winning serves, winning shots, three-point run, missing both serves and unforced errors, etc;

3. This study proposes a more universal and accurate multidimensional momentum chain model. It utilizes a variety of common indicators in tennis matches, and is based on differential equations and the traditional momentum chain model, while introducing the ideas of linear weighting and forgetting curves. Therefore, our model provides a reliable measure of the momentum changes in actual matches.

By applying our model to 6870 international tennis matches from 2011–2023, the prediction accuracy of our model is all above 80%. The results show that the model excludes the interference of unexpected and irrelevant factors in individual matches, and is barely affected by players' personal factors. The model has high prediction accuracy, thus can provide powerful data support for players and coaches to master the match flow and adjust their strategies. Meanwhile, we found that scoring or not is related to the momentum accumulated by both players, which is generated by match-related physiological and psychological reasons. Our

findings not only demonstrate the impact of momentum in breeding success, but also provide a precise reference and basis for quantitative studies of momentum.

## 2. Methods

### 2.1 Design

Tennis is a popular ball sport which is widely played around the world, and the ups and downs of the games are highly entertaining to watch.

In order to better study the changes of momentum in tennis, it is necessary to briefly understand the rules of tennis. There are two main types of matches: singles and doubles, and before the match begins, the side that will serve the ball needs to be determined. Throughout the match, both players take turns to serve the ball. If the server miss the first serve, a second serve shall be made from the same spot, and missing both serves shall result in a point for the opponent. After a point is scored or lost, the next serve shall be made from a different area. The aim of each player is to hit the ball back onto the opponent's court, making it as impossible as possible for the opponent to catch the ball, while ensuring that it does not go out of bounds. The first point of each set is recorded as 15, the second as 30, and the next as 40, and each set must end with at least two more points than the opponent. The first player to win 4 points in each set wins 1 set. A tie-break occurs at the end of the set when both players have 3 points each. After a tie-break at the end of the set, a net win of 2 points wins the set.

Based on its complex rules, tennis match data has many valuable features for the study of momentum or hot hands. In contrast to large team sports such as basketball, a player's behavior is essentially controlled solely by him or herself and influenced only by his or her and the opponent's behavior, with fewer interferences involving other factors (e.g., team strategy, teammate cooperation, etc.), which means players focus their attention on each swing and try their best to score. What's more, since tennis scoring is divided on a ball-by-ball basis, scoring is more independent of each other, unlike in sports such as basketball where there is a greater influence and correlation between scores.

Since the purpose of this study is to investigate the existence and mechanism of the "hot hand effect" in tennis through data analysis, as well as to study the on-court behavior that triggers momentum generation and change, we chose to study the situation of each specific point, including the scorer, the way of scoring, whether it is a serve, whether it is a preferred serve, etc.

### 2.2 Sample

In order to ensure the reliability and objectivity of the data, we consulted the official data provided by the Professional Tennis Federation (https://www.atptour.com/en) to obtain some basic information of professional players in the matches (e.g., breaks of serve, service winners, etc.), and at the same time, we also obtained the specific point data by consulting the statistics and collected data of researchers (https://github.com/JeffSackmann/tennis_slam_pointbypoint), we obtained match details for nearly 6,870 matches from 2011–2023, including 7284 point information in the 2023 Wimbledon Men's Singles Championship.

By finding missing value, testing outliers and other in-depth examination of these data, we found that these data are all true and accurate, which can reflect the actual situation of the players in the game in a more comprehensive way. At the same time, the data contains the game data of the major open tennis tournaments in the past ten years, which can basically cover all kinds of tennis games, providing objective and reliable data support for the study. After the data were properly organized and extracted, we obtained the data set needed for this study. It should be noted that, since the momentum in this study mainly comes from the players

themselves and their opponents, in order to limit the variables and minimize the influence of other factors on the research results, we ignored other uncontrollable and unknown factors, such as the weather, the audience's reactions and so on.

## 2.3 Instruments

In this study, the main data analysis was conducted entirely through computer programming, eschewing the use of traditional experimental apparatus. The code used in this study has been open-sourced in github at the following address: https://github.com/Gsh1111/tennis/tree/main. The details of our research are as follows:

1. Programming Languages: We employed Python as the primary programming language for data manipulation and analysis. This choice was driven by its robust data processing capabilities and extensive library support.

2. Software and Libraries: The analysis relied on several software packages and libraries, including NumPy/Pandas/Matplotlib etc., which facilitated tasks such as numerical operations, data structuring, visualization, and scientific computations.

3. Computational Environment: All computations were performed on a Windows operating system.

4. Reproducibility: To ensure that our findings are reproducible, we have made our code available on GitHub, along with detailed documentation on how to replicate our analysis.

## 2.4 Procedure

The procedure of our study can be divided into several distinct stages, each critical to the development and validation of our model. Here, we detail each step to ensure transparency and reproducibility:

1. Data Collection: As mentioned in Section 2.2, we collected necessary research data and formed the dataset of our study.

2. Model Development: Depending on previous surveys and the match data, we constructed a model that aimed to capture the relationship between various factors and the states of athletes. By detail, we first defined the independent variables and the dependent variable (momentum), then specified the model's structure, including the selection of difference equations and the forgetting curve.

3. Statistical Analysis: With the model established, we proceeded to calculate the specific parameters using statistical methods.

## 2.5 Data analysis

**2.5.1 Game data analysis.** To conduct a preliminary investigation on the match flow, it's essential to visualize the number of points, games and sets won by each player over time. We take the 2023 Wimbledon Men's Singles final as an example to give analysis.

From Fig 1, we can clearly grasp the overall trend of the final. It can be seen that Djokovic won by an overwhelming superiority in the first set. In the second set, a tie-breaker appeared, with Alcaraz scoring the set eventually. In the third set, Alcaraz gained a great advantage at the

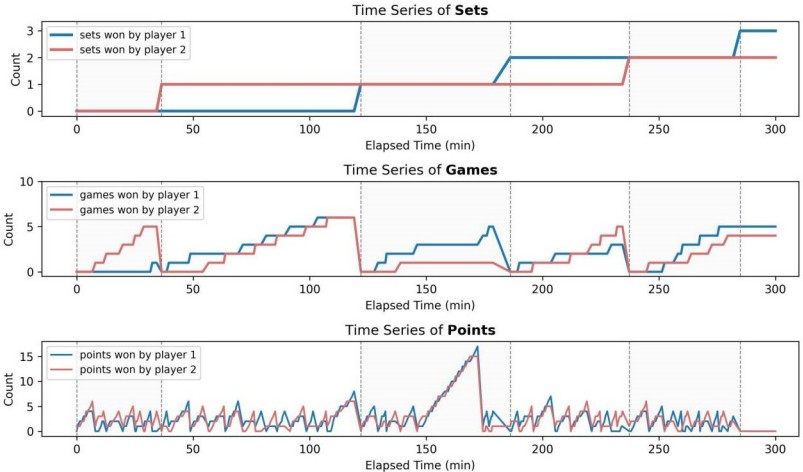

**Fig 1. Time series of points, games and sets.**

set level; for each game, however, points scored by the two players are extremely close, indicating the performance of two players for the third set differed slightly as a matter of fact. The fourth and fifth set both witnessed a reversal of potential winners, where the match grew increasingly intense. After a tortuous battle, Alcaraz achieved the ultimate victory, winning 3 sets out of 5.

**2.5.2 Dependent variables analysis.** Since this study aims to find a way to quantify momentum and investigate the existence and formation mechanism of momentum and the effect of momentum on players' behavior, the dependent variable of the model is momentum. And in further research, the dependent variable is the players' scores, i.e., momentum affects the players' scores of the tournament by influencing their behaviors.

**2.5.3 Independent variables analysis.** Referring to the definition of momentum, unexpected events in a match can randomly affect the mental and physical state of the players, which in turn affects the outcome of the match. Positive events can give players positive psychological cues and may make it easier for them to score in the next round. Since there are many variables in tennis competitions, categorizing and correlating these variables is necessary. To explore the correlation of each index in the original data, we firstly draw the correlation heat map of indexes of each player, as is presented in Figs 2 and 3.

According to Figs 2 and 3, it can be found that the majority of indexes exhibit weak connections, indicating that the correlation between the indexes in the original data is not obvious. Therefore, new indicators need to be formulated so as to better characterize the match.

The Random Forest is a classical bagging model, which utilizes the sample data sets to build a number of decision tree models, and identifies the final classification results based on the votes of classification of these decision tree models. By screening features of existing data sets, we use Random Forest to classify the scorers of each point. Through computation, we find that the accuracy of the model is 0.895. We also draw the ROC curve, as is shown in Fig 4, where AUC equals 0.95, suggesting that the model can accurately predict scorers of subsequent points and capture the corresponding scoring status.

According to the Random Forest model, we obtain several important factors that affect the score, the eight most important factors of which are presented in Fig 5. It can be seen that the points gained by players appears to be an essential factor, so we take the scoring performance of players as the basis of our model. Besides, the importance of some other factors are in the

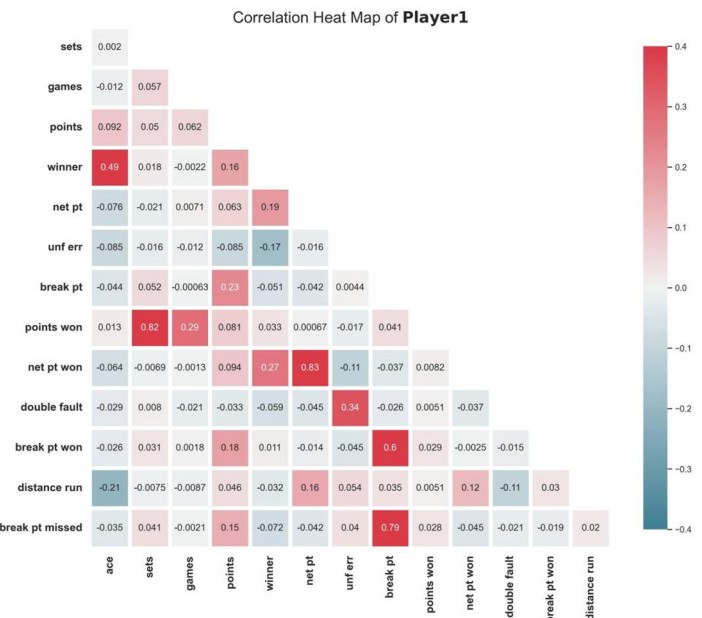

**Fig 2. Correlation heat map of match indexes of player 1.**

forefront, such as unforced errors and untouchable winning shot, so we also introduce them into our model as the specific states of gaining and losing points.

Combining the dataset, rules of tennis and the outcome of the Random Forest algorithm, we summarized the unexpected events that may cause changes in momentum, set them as the independent variables of our model and calculated the magnitude of their impact on victory through statistical analysis methods:

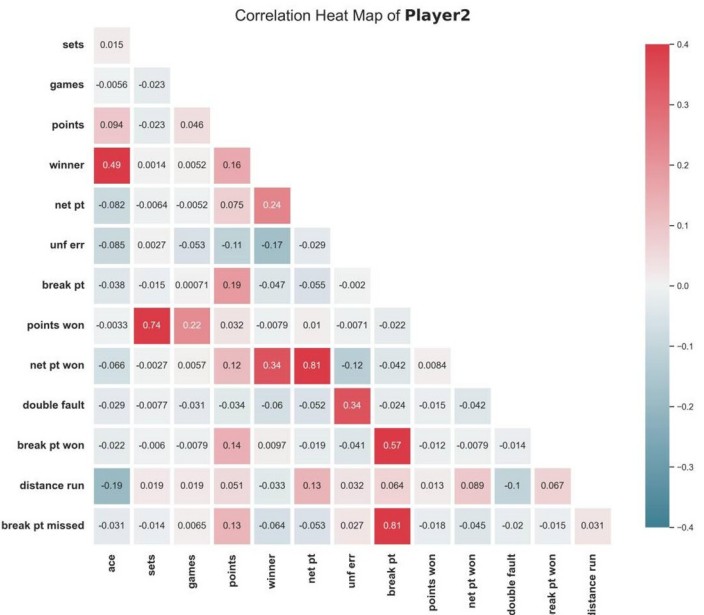

**Fig 3. Correlation heat map of match indexes of player 2.**

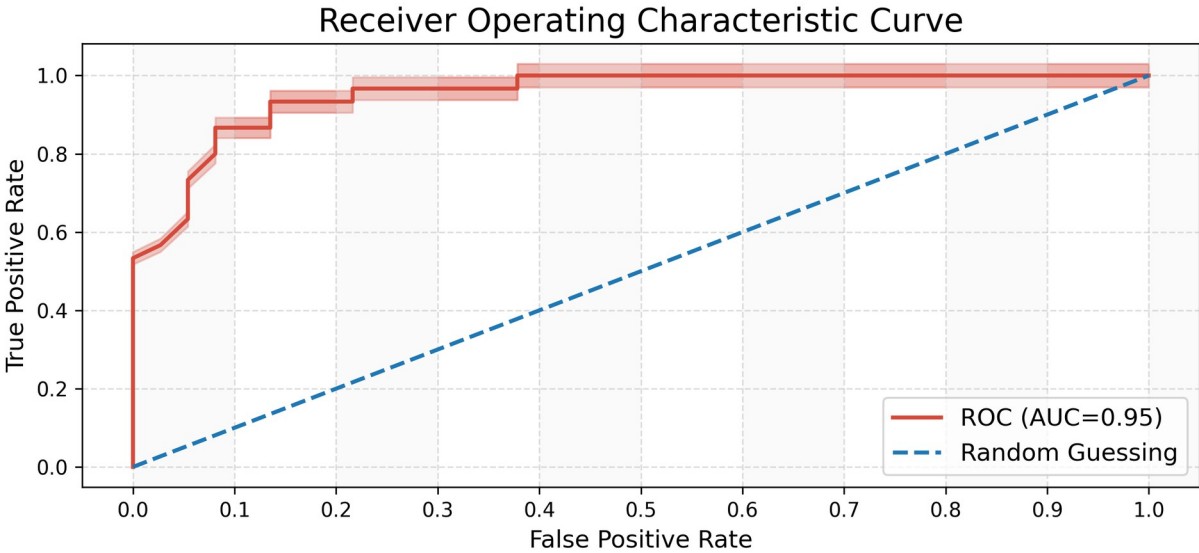

**Fig 4. Receiver operating characteristic curve.**

### 1. Gaining and Losing

There are two scenarios of scoring, namely, scoring by shot (including serve) and scoring by the opponent's unforced errors. In practice, the opponent's unforced errors are little correlated to the scoring player's ability, so we only consider the player's active winning situation. Therein, scoring by serve is barely affected by the opponent's status and can better reflect the player's actual level, which needs to be given particular concern. Next, we need to determine the serve influence weight and the shot influence weight in order to evaluate their respective impacts on scoring a point. Combined with the factor importance obtained from the previous Random Forest model, we identify the serve influence weight and the shot influence weight.

In terms of losing points, we consider unforced errors by the player. As a type of unforced error, the double-fault is a huge loss for the serving side. For this reason, we need to determine the double-fault influence weight and the unforced-error influence weight in order to evaluate their respective impacts on losing a point. Combined with the factor importance obtained from the previous Random Forest model, we identify the double-fault influence weight and the unforced-error influence weight.

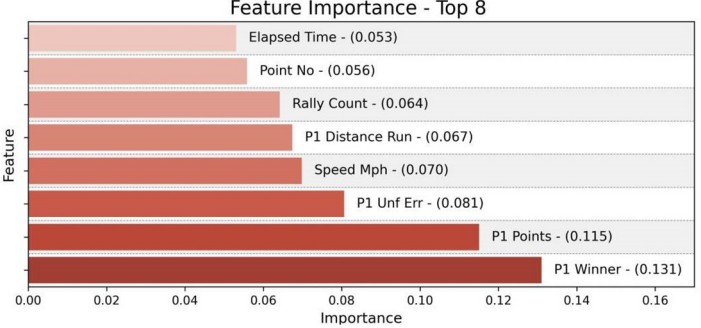

**Fig 5. Feature importance—Top 8 features.**

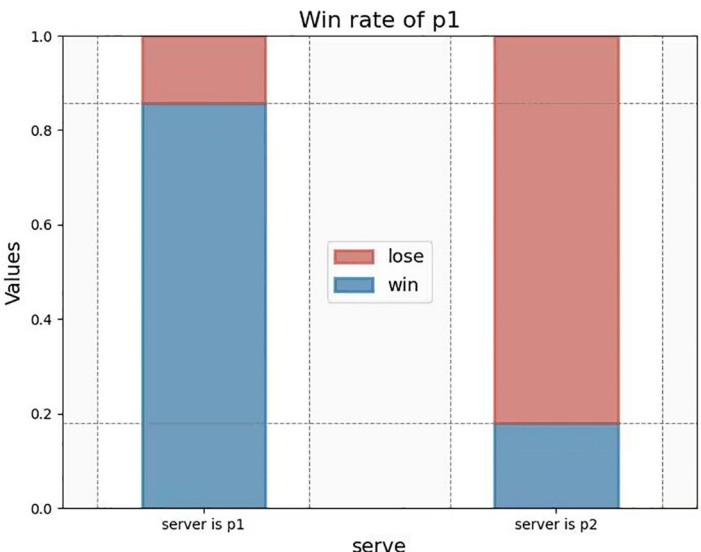

**Fig 6. Serving and winning.**

### 2. The Serving Side

In tennis, the serving side can often gain the initiative by deciding the direction and speed of the serve, hence more likely to win a game. Through the chi-square test on the serving situation and the actual scoring situation, we can see: the Pearson chi-square value is 547.48 and the significance is 0.00 (less than 0.05), indicating that there is a strong correlation between the serving side and the game winner. Through the statistics of entries game_victor and server in the given table, we identify the average probability of the server winning the current game to be 0.84, and the average probability of the server losing the current game to be 0.16, indicating that the server is more likely to score a point (see Fig 6). Accordingly, we determine the serving influence weight.

### 3. Break Points

A break point is a situation in which scoring one more point on the opponent's serve can lead to victory. Thus, it can be used to measure a player's advantage on the opponent's serve. The greater the number of break points, the greater the advantage. On top of this, a break-point victory is a sign that the player is doing even better. Therefore, we introduce the number of break points and the break point score into our model and consider the break-point influence weight.

### 4. Net Shooting

Going to the net is a tennis shooting strategy which assists the player to gain the initiative, reflecting the player's basic skills and reaction speed. Through calculation, the success rate of the player's net shooting is 82%. Meanwhile, since this indicator is not a necessary item for scoring, we adjust its contribution index downward when considering the net influence weight.

### 5. Court Type

Unlike other ball games, there are three different types of tennis courts. In major international tournaments, the Wimbledon Championship is played on grass; the French Open on

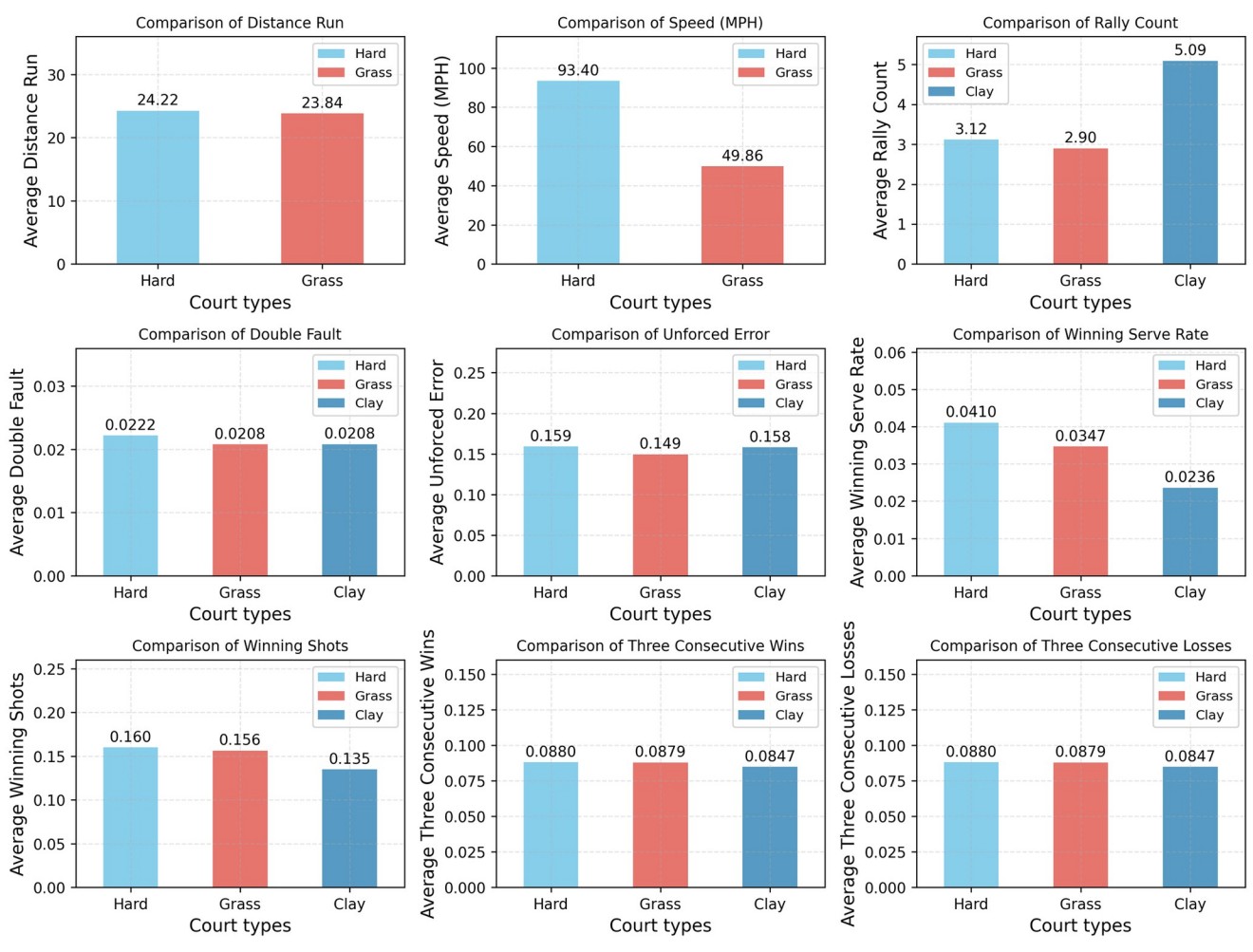

**Fig 7. Indicators in different court types.**

red clay; and the Australian and US Open on hard courts. According to the common sense of sports, different venue materials may have different aspects and degrees of influence on players' sports. In order to ensure the reasonableness of our designed indicators, we chose the 2021 Wimbledon Championship, US Open, and French Open for study, and selected 7,381, 7,486, and 7,146 scoring points with the grass, hard, and clay courts respectively. The scoring points were statistically analyzed to derive the relationship between the court type and the typical indexes of the athletes, as shown in Fig 7. Due to some missing data, we are unable to calculate the running distance and serving speed of players when the court is clay.

For rigorous mathematical analysis, we normalized the above data and calculated the variance of different indicators as shown in Table 1.

**Table 1. Variance of different indicators.**

| Indicator | Distance_Run | Serve_Speed | Rally_Count | Double_Fault | Unforced_Error |
|---|---|---|---|---|---|
| Variance | 0.0026 | 0.1905 | 0.1152 | 0.0010 | 0.0009 |
| Indicator | ACE | Winng Shots | Three Consecutive Wins | Three Consecutive losses | |
| Variance | 0.0928 | 0.0066 | 0.0003 | 0.0003 | |

Combining the variance data and the bar charts, it can be seen that players' double-fault rate, unforced-error rate, winning shots, and three consecutive wins (losses) are relatively stable, with a variance of less than 0.01, whereas players' serve speed and rally counts vary greatly with the type of courts they play on. In particular, players make more hits on average when playing on red clay than on hard or grass courts, and the probability of aces on a clay court is also significantly less than in games on the other two surfaces. This may be due to the fact that the tennis balls subject to greater resistance after contacting the clay, leading to a slower ball speed and higher bounces. So the players have more chances and time to take the initiative to hit the ball and decide the strategy in clay matches, but it is also more difficult to score directly by serving. In addition, players' serve speeds are significantly slower on grass than on hard courts. Similar to the previous analysis, grass is more elastic and has less friction with the tennis ball, resulting in less loss of ball speed, so players do not need to serve at a higher speed to achieve a certain scoring effect, and it is also easier for players to attack their opponents. In contrast, hard court has moderate speed and regular bouncing, which is suitable for players with different playing styles, and the values of indicators tend to be more average.

To conclude, due to the characteristics of court materials, the players' strategies and rhythms during the matches will also be affected. However, it can be seen through the statistical data that the players' double-fault rate, unforced-error rate, aces, winning shots, three consecutive wins, and three consecutive losses are all relatively stable, which are little affected by the court factors, and therefore they can represent the general situation of the players in all types of matches, and are applicable to our model.

Based on the study above, we can screen out the most common characteristics in tennis competitions and divide them into two categories, namely, positive and negative. Positive events include winning serve, winning shots, and scoring three points in a row; negative events include double-faults, unforced errors, and losing three points in a row. Additionally, since the server has a greater chance of winning the game, we take the serving factor into consideration.

Next, we separately count the positive and negative events encountered by the winning and losing players during Wimbledon 2023 Gentlemen's final after the second set, as specified in Table 2. We also use bar charts to visualize the proportion of winners and losers in each event (see Fig 8).

On the whole, the probability of winners experiencing positive events (79.9%) is significantly higher than that of losers experiencing positive events (34%). Through the chi-square test, the chi-square value of negative or positive events experienced by the players is 1183.447, and the significance is 0.00, indicating that events can affect the results of the match by affecting the momentum. Among these events, a three-point run has the biggest impact on the match results.

Meanwhile, by drawing the correlation heat map of events (see Fig 9), it can be seen that the correlation coefficient between similar events is positive, while the correlation coefficient

**Table 2. Positive and negative events encountered by winners and losers.**

| Statistics of Events | Winning | Losing |
|---|---|---|
| p1. Rate of winning serve | 0.10 | 0.04 |
| p2. Number of winning shots | 1893 | 540 |
| p3. Number of three consecutive wins | 1365 | 73 |
| n1. Rate of double-fault | 0.028 | 0.062 |
| n2. Number of unforced error | 810 | 1100 |
| n3. Number of three consecutive losses | 73 | 1365 |

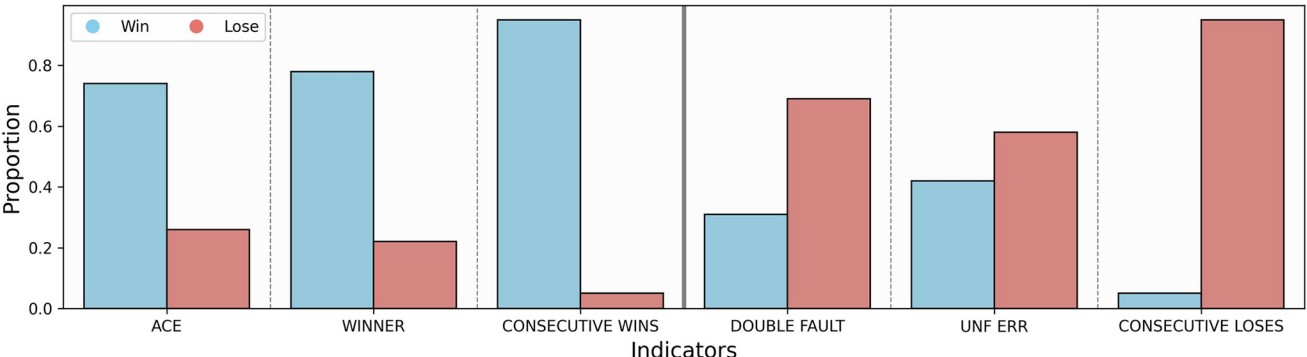

**Fig 8. Positive and negative events encountered by winners and losers.**

between different types of events is negative. This reflects the rationality of the momentum chain, that is, events will have impacts on the subsequent states of the match.

Furthermore, we discover that the average running distance of the winners (13.40m) is shorter than that of the losers (14.47 m), and the number of net-shooting success of the winners (936) was significantly larger than that of the losers (246). This demonstrates that the player with higher momentum has more control and initiative over the match.

To summarize, sudden events will not only affect the psychology and behavior of a player, but also ultimately affect the results of the match. In view of the above evidence, the results of the competition are not random, but intimately linked to momentum. The above factors not only reflect the player's state better, and the test found that they all have different degrees of influence on the subsequent results, so it is reasonable to use these variables as the basis of model construction.

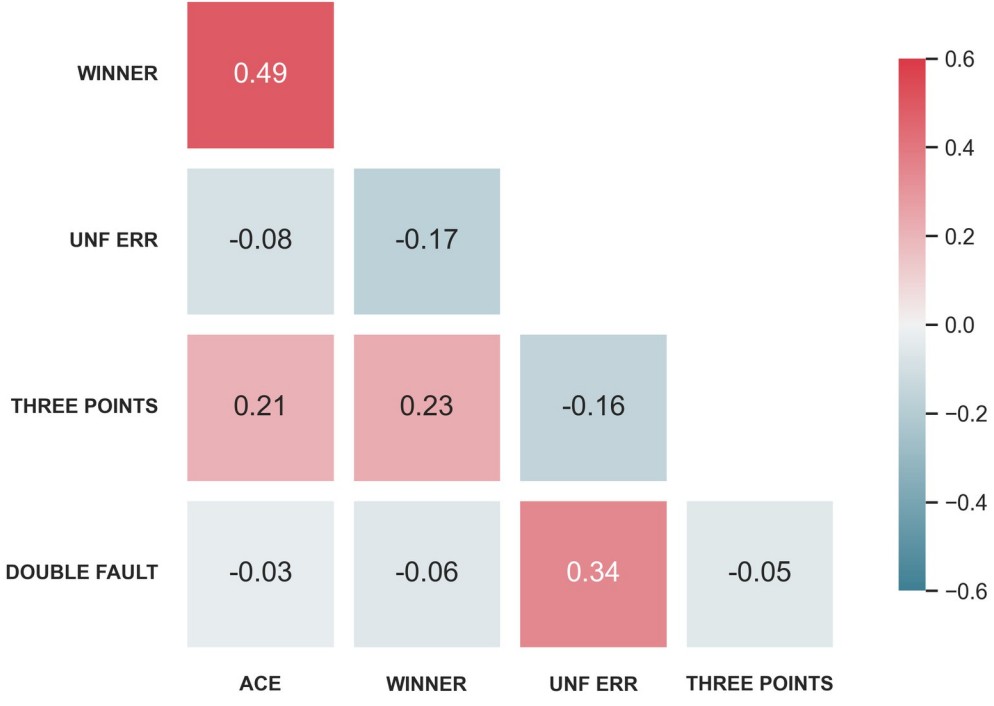

**Fig 9. Correlation heat map of events.**

**2.5.4 Model construction and analysis.** In practice, the underlying momentum chain model can qualitatively observe and explain many changes in the match flows, but it does not provide quantitative indicators and needs to be improved.

Through the statistical analysis of the six indicators included in the four common match events in Section 2.2, we found that they all have different degrees of influence on the players' subsequent scores and other behaviors, so we used these six key indicators (respectively, serve points, stroke points, three consecutive winners, double faults on serve, unforced errors, and three consecutive losers) as the basic variables for calculating the momentum. In the way of combining the variables, we choose to use the linear weighted evaluation model, which is a commonly used quantitative evaluation model. The main idea is to express the higher-level indicators as a linear function of the number of lower-level indicators, and to use the linear function to calculate the score of the higher-level indicators. The score is then used to measure the strengths and weaknesses of the evaluated object. By assigning different weights to the six variables, the different influencing roles of different variables on the magnitude of momentum can be modeled scientifically and clearly.

At the same time, the momentum between different scoring points interacts with each other, i.e., the current momentum is determined by the momentum of the last scoring point and the change in momentum caused by the events during these two scores. Based on this feature, we choose to use the difference equation as the model basis. Difference equations are descriptions of discrete-time systems in which variables at later moments are generated by variables at earlier moments or even previous moments.

In addition, in psychology, the research of German psychologist H. Ebbinghaus proposed the forgetting curve to describe the human brain's law of forgetting new things, which found that forgetting starts immediately after learning, and the process of forgetting is not uniform; initially, the speed of forgetting is very fast, and then it gradually slows down. Although the forgetting curve model proposed by Ebbinghaus was initially only for the memorization and learning of English words, the memory mechanism of the human brain it considers is applicable to many new things. In this study, unexpected events can cause a change in momentum, and the effect this change has on the player's psychology can be understood with the aid of the forgetting curve. Meanwhile, since the time interval between two neighboring score points can be random, and the effect of the previous score point on the next score point decreases with time, which is consistent with the characteristics of human memory for new things, we introduced the difference equation along with a memory factor to measure the effect of the previous momentum on the new momentum after a period of time. Combined with the above discussion, the composition of the MMC model is shown as Fig 10:

We formulate the computational method of momentum as follows:

$$M_i = f \times M_{i-1} + \Delta M \, (M_0 = 0) \tag{1}$$

$$f = 0.6 + 0.4 e^{-\lambda \Delta t} \tag{2}$$

$$\Delta M = \alpha \sum_{k=1}^{3} (w_{pk} \times flag_{pk}) + \beta \sum_{j=1}^{3} (w_{nj} \times flag_{nj}) \tag{3}$$

Where the parameters are standardized in Table 3.

Since the server has a higher chance of scoring, we introduced the serve factor in the model. Based on the contribution of different events to the outcome of the match, we determined the corresponding impact weights, as shown in the table above. In addition to this, at the end of each match, we introduced the impact of the break point factor into the corresponding momentum score in order to better measure the overall performance of the player in a set.

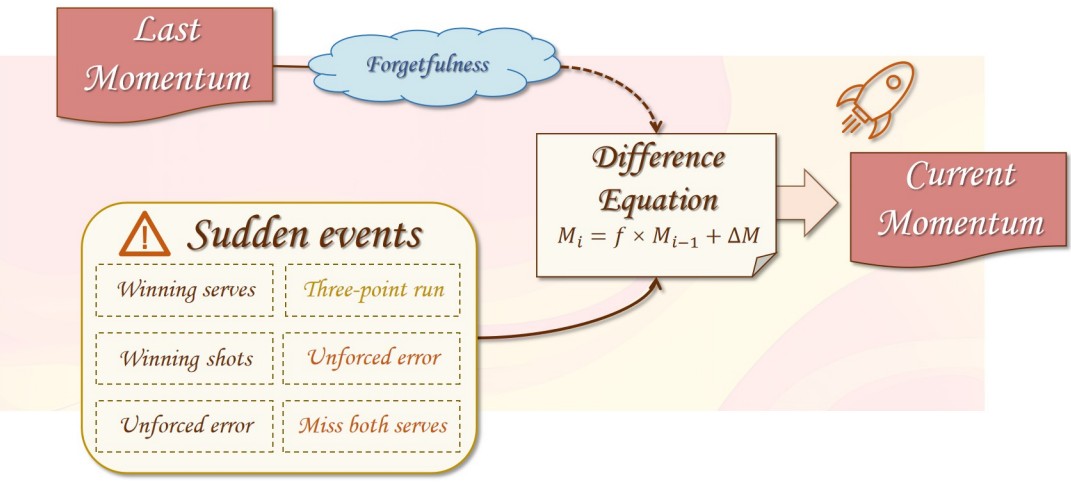

**Fig 10. The composition of the MMC model.**

## 3. Results

By applying our model to the 31 matches of the 2023 Wimbledon men's singles, we calculate the momentum differential between the two sides at the end of each match and predict the winner of the match. The prediction accuracy of the model turns out to be 93.5%, with the predicted winners of two matches "2023-wimbledon-1305" and "2023-wimbledon-1316" different from actual winners. We analyze the reason for the prediction errors and conclude some avenues to improve our model's applicability.

Take match "2023-wimbledon-1305" as an example (see Fig 11):

A closer look at the momentum scores of the two shows that Daniil Medvedev's momentum in the final key stage has declined significantly, which is inconsistent with the actual performance. The reason behind this strange phenomenon is that there have been many three-point losses in a row. By adjusting the weight of the impact of three-points runs on the results, we get the correct prediction results. In addition, we note that both prediction errors occurred in the third round of the tournament, that is, in the 32-strong competition. Therefore, we may need to adjust the weight of specific parameters for different stages of the tournament to fit the characteristics of each round of the tournament in the future.

In summary, when we use our model to predict the result of a certain match, it is not enough to only consider the factors that affect the outcome of a game. Based on the complex rules of tennis matches, the weights of each part need to be adjusted to suit different stages, in order to improve the prediction accuracy of the model for the entire match results.

**Table 3. Positive and negative events encountered by winners and losers.**

| Symbol | Description | Value |
|---|---|---|
| $M_i$ | the momentum at score point i | \ |
| $\Delta M$ | the variation of momentum | \ |
| $f$ | the forgetting coefficient | $0.6 + 0.4e^{-\lambda \Delta t}$ |
| $\alpha$ | the positive influence weight | server:1.2, else:1 |
| $\beta$ | the negative influence weight | server:1, else:1.2 |
| $w_{pk}$ ($k = 1,2,3$) | the influence weight of each positive event | $w_{p1} = 0.30, w_{p2} = 0.32, w_{p3} = 0.38$ |
| $w_{nj}$ ($j = 1,2,3$) | the influence weight of each negative event | $w_{n1} = 0.32, w_{n2} = 0.26, w_{n3} = 0.42$ |

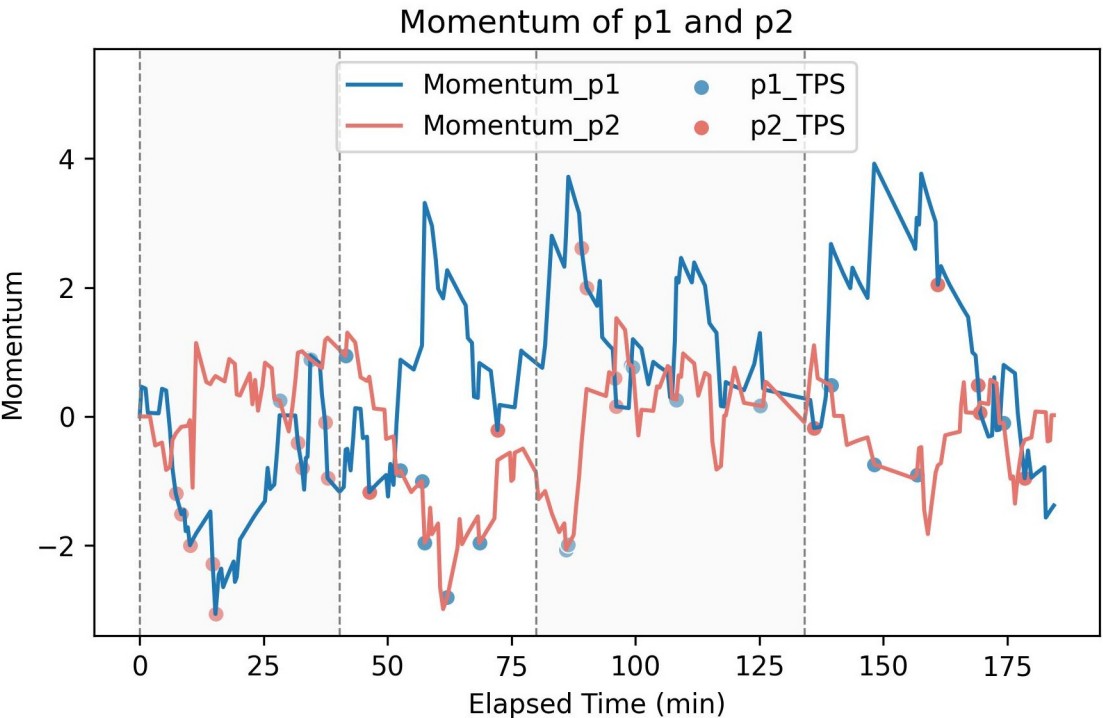

**Fig 11. Momentum score of players in 2023-wimbledon-1305.**

Additionally, We collect data of the tennis tournament from 2011 to 2023 (including the Wimbledon Champion, French Open, US Open and Australian Open), and employ our model to predict the result of each tournament. The information and prediction accuracy of each tournament are specified in Table 4:

From Table 3, we can see that our model can apply to nearly all of these tennis tournaments. However, the accuracy tends to be influenced by the type of the tournament. Therein, the prediction accuracy of the men 's singles is the highest, and the prediction accuracy of the mixed doubles and women's singles is not good enough. Meanwhile, the prediction results are little affected by the competition location and land type, indicating that our model has strong versatility. To improve the applicability of the model to different tournaments, we consider adding new indicators according to the rules of the competition, and adjusting the weight of the indicators to adapt to the characteristics of various tournaments.

**Table 4. Information and prediction accuracies of other tournaments.**

| Tournament type | Land type | Number of matches analyzed | Average accuracy | Location |
|---|---|---|---|---|
| **Men's singles (2023)** | grass | 31 | 93.50% | Wimbledon |
| **Women's singles (2023)** | grass | 127 | 85.04% | Wimbledon |
| **Mixed doubles** | grass | 138 | 80.37% | Wimbledon |
| **Men's doubles** | grass | 496 | 84.40% | Wimbledon |
| **Men's singles** | hard | 2633 | 90.14% | United States of America |
| **Men's doubles** | hard | 534 | 92% | United States of America |
| **Mixed doubles** | hard | 119 | 63.32% | United States of America |
| **Men's singles** | hard | 1360 | 92.81% | Commonwealth of Australia |
| **Men's singles** | clay | 1432 | 89.12% | French Republic |

Moreover, for other types of sport competitions, we need to adjust the parameters in the formulas of our model. Since the variables introduced in our model are highly correlated with tennis matches (e.g., double-faults and winning shots), they are less versatile for other sports. However, for different sports, the above MMC model can still be used by adding competition characteristic indicators, such as racket techniques in table tennis competitions and forehand and backhand catch in badminton competitions. In this way, our model is still able to quantify player performances and predict the swings in sport competitions.

## 4. Discussion

In this paper, we investigate the effects of typical positive or negative behavioral events on the performance of professional athletes of comparable strength in tennis and construct a multidimensional momentum chain model based on differential equations through a variety of statistical methods. We find that players who experience more positive events can have a higher probability of scoring points or even winning, which suggests that there is a significant hot hand phenomenon in tennis, and that there is a significant effect of momentum on tennis.

We find that, overall, the probability of a winner experiencing a positive event (79.9%) is significantly higher than the probability of a loser experiencing a positive event (34%); for the specific variables, the probability of a server winning the game (84%) is much higher than that of a non-server (16%), and the probability of going on to score after winning three points in a row is much higher than that of the opponents. At the same time, the running distance of the scorer is significantly shorter than that of the scorer who loses the points, and the success rate of scoring on the net is also higher, suggesting that positive outcomes such as winning points can also allow players to better master the match flows and thus create appropriate scoring opportunities for themselves. In terms of negative events, winners also had a lower rate of double faults on serves (2.8%) than losers (6.2%), and fewer consecutive three-point losses and unforced errors, which also shows that experiencing fewer negative events is a strong contributing factor to winning from a whole-match perspective.

In addition, in order to minimize the influence of other uncontrolled factors, we have controlled for the strength gap between the players in our study to ensure that our findings are robust and general, and thus to demonstrate that the influence and mechanism of momentum and hot hand effects are noteworthy and strategically important in all types of tennis matches. The starting point of this study is the men's singles final at Wimbledon, where the difference between the two players in the world rankings is very small, and there has not been any significant overwhelming victory in previous matches, so it can be assumed that the difference in strength between the two players is not a major influence factor. At the same time, the other matches involved in the study were later rounds of major international tournaments, where the strength gap between the players was relatively small. This provides solid visual evidence of the "success breeds success" theory, which suggests that scoring and winning are not entirely determined by luck or ability differentials, but are influenced by the hot hand effect. It is worth mentioning that our study considered and controlled the time factor, and introduced the forgetting curve to standardize the model parameters in order to simulate the athletes' psychological response to momentum changes in real-life situations.

In the introduction, it was mentioned that the "hot hand effect" was first used to describe the game state of a basketball player scoring consecutive points, and it has met with many rebuttals at the time of its introduction, with research suggesting that many spectators' and even athletes' perceptions of, and responses to, momentum (the hot hand effect) stems from a bias toward serial memory rather than memory for change [10]. Gilovich, Vallone et al. [23] strongly argued that biased judgments of chance events (rather than biased memory) are

associated with beliefs about the law of small numbers [30], i.e., the general consciousness overstates the breadth of the impact of such effects [31]. However, our findings challenge this view by analyzing experimental data that the hot hand phenomenon is significant in tennis and that the psychological and physiological changes in athletes are not biased against serial memory.

Furthermore, our study provides strong evidence support for the generation and mechanism of action of momentum. The study by Taylor and Demick [5] proposed a multidimensional momentum chain model that qualitatively describes the mechanism of momentum, which considers the contingencies of oneself and the opponent to be the cause of the generation of momentum, and at the same time, the momentum can cause cognitive, affective, physiological, and behavioral changes, which in turn affects the probability of success in the next step. Based on this, our study found the typical contingencies in tennis matches, and used the results of statistical analysis to quantify the different degrees of influence that various types of time have on athletes. Meanwhile, the athletes also showed different reactions psychologically and behaviorally after experiencing the unexpected events, which is consistent with the above study.

From a psychological perspective, the results of this study support the theory of self-efficacy [32]. Self-efficacy is a strong predictor of athletic performance [33] and has a direct and independent effect on success [34]. At certain stages of the game, tennis players experience positive psychological cues after positive events, which is consistent with the concept of psychological momentum [10]. The psychological momentum is defined as a psychological advantage, and the activation of psychological momentum will lead to further success. Although this study did not directly address the psychological state of the players, it was found through analysis that three consecutive points and three consecutive points lost had the greatest impact on the players' momentum, and these types of consecutive positive and negative events have been shown to be one of the most essential triggers of the hot hand phenomenon, thus this study also provides indirect evidence to support the self-efficacy theory.

From a physiological perspective, by applying our model to various types of tennis matches, it can be found that its prediction accuracy for men's singles matches is significantly higher than that for mixed doubles and women's singles matches, a result that supports the hypothesis that the winner's effect is mediated by an androgenic feedback loop. In this theory, endogenous secretion of testosterone is the main mediator of momentum production and action, and high levels of testosterone contribute to dominant nonverbal communication while Low levels of testosterone inhibit this behavior. Winning increases testosterone levels, which can increase the likelihood of further wins [35], however, we cannot be certain of this explanation as we did not further investigate other non-androgenic mechanisms mediating the winner effect. In the present study, female athletes also presented a confident, exuberant state after experiencing a positive event, such as upright posture and strutting, whereas they also tended to present an avoidant, remorseful state after experiencing a negative event, such as a stooped posture and a disgusted gaze. Thus, differences in prediction accuracy may arise because circulating testosterone levels in women are on average about 10% of those in men, resulting in a less significant effect of momentum for women.

Different from the traditional qualitative analysis, we were inspired by the quantitative calculation methods in archery [29], and proposed this multidimensional momentum chain model to find a way to quantify the effect of momentum on players more accurately as well as to predict the change of the game situation. Compared with other momentum studies, the advantages of this model are as follows:

1. Scientific selection of indicators. We identified the most important factors affecting scoring and used statistical methods to identify key events affecting the flow of the game. These factors are highly relevant to the game and can scientifically and comprehensively reflect the game situation and athletes' status.

2. Comprehensive and practical. In measuring players' performance, we use scoring time points to divide the game and quantify players' performance. When measuring a player's momentum, we comprehensively consider the impact of the momentum chain on all aspects of the player.

3. Clarity and intuition. The multidimensional momentum chain model we built is simple to calculate and the results are clear.

4. High accuracy. We applied our model to a variety of tennis matches and found that the prediction accuracy was almost always above 80%, indicating that our model is stable and accurate and applicable to most matches in tennis.

Unfortunately, due to the uncertainties associated with human competition, our model has some drawbacks. Firstly, it requires high data quality, and when some indicators are lacking in the dataset, our model may not be able to provide the best predictive results. In the further promotion, we will continue to search for more detailed associations between indicators to formulate more general, accessible and observable variables to form the basis of the model and provide simpler strategies for athletes and coaches. Secondly, the indicators involved in this model are unique to tennis, making it difficult to obtain reliable predictions when applying it to other competitions. However, in the study of other similar games, we can choose appropriate variables according to the characteristics of different games, and construct a universal model suitable for multiple sports based on the idea of multidimensional momentum chain. Finally, although our study provides evidence for the momentum mechanism in tennis, the effect of momentum on different players has not been carefully calculated, and their responses to the same momentum need to be further considered. In addition, this study focuses on the "hot hand effect" in tennis, and less on the effects of momentum in other sports. Momentum itself is a relatively abstract concept, and there are many controversial and doubtful issues, and its generation and influence mechanisms need further study.

To sum up, this study first demonstrated the existence of the "hot hand effect" in tennis, and discovered the mechanism of momentum generation and its psychological and physiological effects; on the other hand, it then provided an accurate and reliable quantitative model of momentum, the multidimensional momentum chain model, which provides reliable suggestions for athletes and coaches to better master momentum. Meanwhile, this study also provides references and ideas for the study of momentum in other sports as well as other fields.

## 5. Conclusions and practical applications

In this paper, we study the effects of common events in tennis matches on the performance of professional tennis players. By examining changes in the behavior of players in international tournaments, we find that players who experience more positive events have a higher probability of scoring points and even winning, and that they also present higher energy and will be more proactive in their game strategy. This suggests that there is a significant hot hand phenomenon in tennis, i.e., the experience of success increases the likelihood of subsequent success, and that this phenomenon is closely related to the psychological state and physiological responses of players.

At the same time, this study innovatively proposed this Multidimensional Momentum Chain model to find a way to more accurately quantify the effect of momentum on players and to predict changes in the match flows. This model is empirically validated with very reliable results. The model and the results of the study revealed that the athlete's state is closely related to the events on the field of play, and that athletes or coaches can take active measures to promote the growth of the athlete's momentum according to the situation on the game field. This finding is important for understanding momentum dynamics in sport competitions, and also provides a basis and ideas for understanding, predicting, and adjusting athlete performance in high-pressure environments.

Applying our model to practical use, it is suitable for understanding and predicting real-time situation changes in the playing field. Based on our model and the results of the study, the following recommendations are proposed for tennis players:

1. Formulate targeted strategies based on specific match stages: The overall momentum can reflect the general state of the players. If the momentum is basically above zero, it indicates that the player is in good condition. Before starting a new match, players can compare the historical momentum of themselves and their opponents to find out the certain stage where they usually take the lead. This can help them make timely adjustments and win advantages over their opponents.

2. Improve psychological quality and the ability to confront emergencies: The frequency of momentum fluctuation can reflect the ability of players to cope with sudden events. Fierce fluctuation shows that the player is susceptible to emergencies and lacks self-regulation ability. Therefore, players can take actions to improve their psychological quality in peacetime training and neutralize the impact of emergencies on their performances.

3. Explore the factors with the biggest influence on themselves: Players can analyze their historical data to find out the factors that have the greatest impact on themselves according to their previous momentum scores. During a new match, they can pay special attention to those factors, and make timely intervention and adjustment.

Considering that coaching is an integral and very important part of the tennis game, we have also provided tennis coaches with the following suggestions based on the model to help their players better utilize and grasp the flow of the court:

1. Take timely measures to cope with sudden events: Our model classifies many common events during the match, when one of those events occur, you can take some specific actions (such as tactical suspension) to weaken the impact of negative events and strengthen the impact of positive events.

2. Use momentum curves to identify key influencing factors: You can use our model to review the status of players in historical games, and then analyze the events that have the greatest impact on players' performance. During regular training sessions, you can develop specific training plans for players that target these events. For example, if a player's momentum is significantly influenced by their serving mistakes, then the focus of the training plan could be to improve their serving skills. This can help avoid the occurrence of negative events, and increase players' self-regulation ability.

3. Adjust tactics based on opponents' characteristics: When facing a new opponent, you can use our model to evaluate the opponent's strength and weaknesses based on their momentum curve. This may help you to evaluate and predict the match flows. During the match,

you can also take targeted actions to stop opponents from gaining a higher momentum and help your players to win advantage.

## Acknowledgments

We would like to extend our sincere thanks to Professor Yurong Sun for her guidance on the initial revision and typesetting of our manuscript. Her expertise and meticulous attention to detail have been invaluable to the improvement of our paper. We are also grateful to the tennis enthusiasts who diligently collected the match data, providing our research with a robust and rigorous empirical foundation. Furthermore, our appreciation goes to the tennis players and coaches for their passion and dedication to the sport. Their commitment and perseverance exemplify the pursuit of excellence embodied by the Olympic spirit, which serves as an inspiration to us all.

This work would not have been possible without their contributions and support.

## Author Contributions

**Conceptualization:** Jingya Wang, Yuanyun Zhou.

**Data curation:** Jingya Wang, Sihang Guo.

**Formal analysis:** Jingya Wang, Sihang Guo, Yuanyun Zhou.

**Investigation:** Jingya Wang, Sihang Guo, Yuanyun Zhou.

**Methodology:** Jingya Wang, Yuanyun Zhou.

**Project administration:** Jingya Wang.

**Resources:** Jingya Wang, Sihang Guo.

**Software:** Sihang Guo.

**Supervision:** Jingya Wang.

**Validation:** Sihang Guo.

**Visualization:** Jingya Wang, Sihang Guo, Yuanyun Zhou.

**Writing – original draft:** Jingya Wang.

**Writing – review & editing:** Jingya Wang, Yuanyun Zhou.

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
