## [Decision Letter · Decision Letter 0]

26 Nov 2024

PONE-D-24-45252A Multidimensional Momentum Chain Model for Tennis Matches Based on Difference EquationsPLOS ONE

Dear Dr. Wang,

Thank you for submitting your manuscript to PLOS ONE. After careful consideration, we feel that it has merit but does not fully meet PLOS ONE’s publication criteria as it currently stands. Therefore, we invite you to submit a revised version of the manuscript that addresses the points raised during the review process.

We look forward to receiving your revised manuscript.

Kind regards,

Gustavo De Conti Teixeira Costa, Ph.D

Academic Editor

PLOS ONE

Journal Requirements:

2. Please note that PLOS ONE has specific guidelines on code sharing for submissions in which author-generated code underpins the findings in the manuscript. In these cases, all author-generated code must be made available without restrictions upon publication of the work. 

Please review our guidelines at https://journals.plos.org/plosone/s/materials-and-software-sharing#loc-sharing-code and ensure that your code is shared in a way that follows best practice and facilitates reproducibility and reuse.

3. Please note that your Data Availability Statement is currently missing the repository name and the DOI/accession number of each dataset (https://www.atptour.com/en). If your manuscript is accepted for publication, you will be asked to provide these details on a very short timeline. We therefore suggest that you provide this information now, though we will not hold up the peer review process if you are unable.

**Additional Editor Comments: **

Dear authors,

Based on the peer review, I kindly ask you to meet the demands presented by the reviewers, highlighting:

Identify and delimit the sections and subsections of the study, such as: Introduction; Method (Design, Sample, Instruments, Procedure, Data Analysis); Results; Discussion; Conclusions and Practical Applications.

In addition, facilitate the understanding of the data, restricting or indicating its applicability according to the type of tennis playing ground. I await the next revised version.

Best regards,

Gustavo De Conti

Reviewers' comments:

Reviewer's Responses to Questions

**Comments to the Author**

1. Is the manuscript technically sound, and do the data support the conclusions?

Reviewer #1: Yes

Reviewer #2: Yes

2. Has the statistical analysis been performed appropriately and rigorously? 

Reviewer #1: N/A

Reviewer #2: Yes

3. Have the authors made all data underlying the findings in their manuscript fully available?

Reviewer #1: Yes

Reviewer #2: Yes

4. Is the manuscript presented in an intelligible fashion and written in standard English?

Reviewer #1: Yes

Reviewer #2: Yes

5. Review Comments to the Author

Reviewer #1: The paper presents accurate information related to tennis, but it needs to better structure and specify all sections according to the guidelines provided by the journal. The sections and subsections of the study are:

Introduction; Method (Design, Sample, Instruments, Procedure, Data Analysis); Results; Discussion; Conclusions and

Practical Applications.

Keep in mind that there is a lot of information that needs to be specified and placed in the appropriate section.

Reviewer #2: 1. Whether the selection of video materials excluded the influence of external variables such as weather and court types (grass, hard, clay). These factors may significantly affect players' psychological and behavioral performance, thereby indirectly influencing momentum. For instance, grass courts may lead to faster-paced matches, while clay courts emphasize endurance and strategy, potentially impacting the formation and maintenance of momentum differently.

2. Whether the significance levels were annotated following academic standards. It is suggested to include bar charts to highlight significant differences and ensure that the formatting of figures, tables, and references complies with the requirements of the journal.

6. PLOS authors have the option to publish the peer review history of their article (what does this mean?). If published, this will include your full peer review and any attached files.

Reviewer #1: No

Reviewer #2: No

---

## [Author Response · Author response to Decision Letter 0]

10 Dec 2024

Academic Editor: Identify and delimit the sections and subsections of the study, such as: Introduction; Method (Design, Sample, Instruments, Procedure, Data Analysis); Results; Discussion; Conclusions and Practical Applications.

In addition, facilitate the understanding of the data, restricting or indicating its applicability according to the type of tennis playing ground. I await the next revised version.

The authors’ answer: Thank you for your suggestions and for your attention to our manuscript. We have made adjustments to the structure of our manuscript in accordance with the recommendations from you and the reviewers, and it now complies with the requirements of PLOS ONE. The specific revisions are highlighted in red. Additionally, we have included further experiments to enhance the scientific rigor of our study. The content of these experiments, as well as the conclusions drawn from them, have been added to the manuscript in blue. Please find the updated manuscript attached. Furthermore, we have addressed the concerns you raised regarding the accessibility of our data and code. We have made improvements in this area and have included accessible links for both. All the data and code involved in our study can be viewed in the github project at the following link：https://github.com/Gsh1111/tennis/tree/main. Thank you again for your concern. We hope that our manuscript now meets the publication criteria of PLOS ONE.

Reviewer #1: The paper needs to better structure and specify all sections according to the guidelines provided by the journal. The sections and subsections of the study are:

Introduction; Method (Design, Sample, Instruments, Procedure, Data Analysis); Results; Discussion; Conclusions and Practical Applications.

Keep in mind that there is a lot of information that needs to be specified and placed in the appropriate section.

The authors’ answer: We would like to extend our heartfelt gratitude for your valuable suggestions and reminders. Upon reflection, we acknowledge that our initial submission did indeed fall short in terms of formatting, which led to a lack of clarity and logical organization in our manuscript. We sincerely apologize for any inconvenience this may have caused.

In response to your feedback, we have diligently revised the order of various sections in our paper and have made additions, deletions, and modifications to certain content. Our manuscript now strictly adheres to the formatting requirements you provided. The specific sections that have been amended are highlighted in red within the revised manuscript for your review. We hope the revised manuscript could be acceptable for you. It is worth mentioning that we have made major changes in the Instruments and Procedure sections, which are excerpted below for your review (it can also be found in lines 226-253 of the revised manuscript)

“2.3 Instruments

In this study, the main data analysis was conducted entirely through computer programming, eschewing the use of traditional experimental apparatus. The code used in this study has been open-sourced in github at the following address: https://github.com/Gsh1111/tennis/tree/main. The details of our research are as follows:

1.Programming Languages: We employed Python as the primary programming language for data manipulation and analysis. This choice was driven by its robust data processing capabilities and extensive library support.

2.Software and Libraries: The analysis relied on several software packages and libraries, including NumPy/Pandas/Matplotlib etc., which facilitated tasks such as numerical operations, data structuring, visualization, and scientific computations.

3.Computational Environment: All computations were performed on a Windows operating system.

4.Reproducibility: To ensure that our findings are reproducible, we have made our code available on GitHub, along with detailed documentation on how to replicate our analysis.

2.4 Procedure

The procedure of our study can be divided into several distinct stages, each critical to the development and validation of our model. Here, we detail each step to ensure transparency and reproducibility:

1.Data Collection: As mentioned in Section 2.2, we collected necessary research data and formed the dataset of our study. 

2.Model Development: Depending on previous surveys and the match data, we constructed a model that aimed to capture the relationship between various factors and the states of athletes. By detail, we first defined the independent variables and the dependent variable (momentum), then specified the model's structure, including the selection of difference equations and the forgetting curve.

3.Statistical Analysis: With the model established, we proceeded to calculate the specific parameters using statistical methods.”

Reviewer #2: 1. Whether the selection of video materials excluded the influence of external variables such as weather and court types (grass, hard, clay). These factors may significantly affect players' psychological and behavioral performance, thereby indirectly influencing momentum. For instance, grass courts may lead to faster-paced matches, while clay courts emphasize endurance and strategy, potentially impacting the formation and maintenance of momentum differently.

The authors’ answer: We are very grateful for your professional and constructive advice. The factors you mentioned are indeed worthy of investigation and have provided us with significant insights.

Firstly, regarding the weather factor, we had taken it into consideration before conducting our data analysis. However, the event organizers often strive to minimize the impact of adverse weather conditions on the fairness of the competition, which is why the matches are almost always held under clear skies. Additionally, the match data we studied did not record the weather conditions at the time, leading us to omit the weather factor in our momentum model. We have added an explanation of this in Section 2.2 (in lines 224-225) of our paper, highlighted in blue for your reference.

Secondly, concerning the court types, your suggestion is highly enlightening. Following your advice, we conducted a more detailed study of the men's singles match data across different venues (grass, hard, clay), including but not limited to comparing running distances, service point win rates, and double fault rates. The study results were added in the fifth point for of section 2.5.3 of the article, specifically in lines 342-377 of the revised manuscript. Our research revealed that the venue has varying degrees and aspects of influence on momentum in tennis matches. We have added more detailed conclusions in the second part of our paper, also highlighted in blue for your review.

Please find the revised sections in the attached manuscript. 

For your convenience, we have extracted some of the main research findings for your reference, as shown below:

“Combining the variance data and the bar charts, it can be seen that players' double-fault rate, unforced-error rate, winning shots, and three consecutive wins (losses) are relatively stable, with a variance of less than 0.01, whereas players' serve speed and rally counts vary greatly with the type of courts they play on. In particular, players make more hits on average when playing on red clay than on hard or grass courts, and the probability of aces on a clay court is also significantly less than in games on the other two surfaces. This may be due to the fact that the tennis balls subject to greater resistance after contacting the clay, leading to a slower ball speed and higher bounces. So the players have more chances and time to take the initiative to hit the ball and decide the strategy in clay matches, but it is also more difficult to score directly by serving. In addition, players' serve speeds are significantly slower on grass than on hard courts. Similar to the previous analysis, grass is more elastic and has less friction with the tennis ball, resulting in less loss of ball speed, so players do not need to serve at a higher speed to achieve a certain scoring effect, and it is also easier for players to attack their opponents. In contrast, hard court has moderate speed and regular bouncing, which is suitable for players with different playing styles, and the values of indicators tend to be more average.

To conclude, due to the characteristics of court materials, the players' strategies and rhythms during the matches will also be affected. However, it can be seen through the statistical data that the players' double-fault rate, unforced-error rate, aces, winning shots, three consecutive wins, and three consecutive losses are all relatively stable, which are little affected by the court factors, and therefore they can represent the general situation of the players in all types of matches, and are applicable to our model.”

Reviewer #2: 2. Whether the significance levels were annotated following academic standards. It is suggested to include bar charts to highlight significant differences and ensure that the formatting of figures, tables, and references complies with the requirements of the journal.

The authors’ answer: We would like to express our sincere gratitude for your suggestions, which have been immensely helpful in enhancing the rigor and clarity of our research. Following your reminders, we have reviewed all of our data analysis conclusions and visualizations. The majority of our data analyses have provided significance analysis that meets academic standards. For the few instances where there might have been omissions, we have supplemented with additional visual evidence, such as bar charts, and other necessary significance analysis data. These revisions were primarily focused on the research concerning the court types, in the fifth point for of section 2.5.3 of the article, specifically in lines 342-377 of the revised manuscript. They were highlighted in blue for your review. Please find the updated sections in the attached revised manuscript. Thank you for your valuable feedback and kind consideration.

---

## [Editor Report · Decision Letter 1]

13 Dec 2024

A Multidimensional Momentum Chain Model for Tennis Matches Based on Difference Equations

PONE-D-24-45252R1

Dear Dr. Wang,

We’re pleased to inform you that your manuscript has been judged scientifically suitable for publication and will be formally accepted for publication once it meets all outstanding technical requirements.

Kind regards,

Gustavo De Conti Teixeira Costa, Ph.D

Academic Editor

PLOS ONE
---

## [Editor Report · Acceptance letter]

17 Dec 2024

PONE-D-24-45252R1 

PLOS ONE

Dear Dr. Wang, 

I'm pleased to inform you that your manuscript has been deemed suitable for publication in PLOS ONE. Congratulations! Your manuscript is now being handed over to our production team.

Kind regards, 

on behalf of

Dr. Gustavo De Conti Teixeira Costa 

Academic Editor

PLOS ONE